# Achieving ZT = 2.2 with Bi-doped n-type SnSe single crystals

Anh Tuan Duong[1], Van Quang Nguyen[1], Ganbat Duvjir[1], Van Thiet Duong[1], Suyong Kwon[2], Jae Yong Song[2], Jae Ki Lee[3], Ji Eun Lee[3], SuDong Park[3], Taewon Min[4], Jaekwang Lee[4], Jungdae Kim[1] & Sunglae Cho[1]

Recently SnSe, a layered chalcogenide material, has attracted a great deal of attention for its excellent p-type thermoelectric property showing a remarkable ZT value of 2.6 at 923 K. For thermoelectric device applications, it is necessary to have n-type materials with comparable ZT value. Here, we report that n-type SnSe single crystals were successfully synthesized by substituting Bi at Sn sites. In addition, it was found that the carrier concentration increases with Bi content, which has a great influence on the thermoelectric properties of n-type SnSe single crystals. Indeed, we achieved the maximum ZT value of 2.2 along $b$ axis at 733 K in the most highly doped n-type SnSe with a carrier density of $-2.1 \times 10^{19}$ cm$^{-3}$ at 773 K.

[1] Department of Physics and Energy Harvest-Storage Research Center, University of Ulsan, Ulsan 680-749, Republic of Korea. [2] Division of Industrial Metrology, Korea Research Institute of Standards and Science (KRISS), Daejeon 305-340, Republic of Korea. [3] Thermoelectric Conversion Research Center, Creative and Fundamental Research Division, Korea Electrotechnology Research Institute (KERI), Changwon 51543, Republic of Korea. [4] Department of Physics, Pusan National University, Busan 605-735, Republic of Korea. Correspondence and requests for materials should be addressed to S.C. (email: slcho@ulsan.ac.kr).

The performance of a thermoelectric material is estimated via the relation of the Seebeck coefficient ($S$), electrical conductivity ($\sigma$) and thermal conductivity ($\kappa$) at a temperature ($T$), which is called the thermoelectric figure of merit, $ZT = S^2 \sigma T / \kappa$. To utilize a material's thermoelectric properties for various applications, observation of a ZT value above 1 is a critical goal of the thermoelectric community. Research aimed at increasing ZT values in recent decades has been focused on the following three areas. First, the Seebeck coefficient can be increased through appropriate carrier doping or energy filtering of charge carriers[1,2]. Second, lowering the effective mass of the carriers and modulation doping in a quantum well can enhance the mobility ($\mu$) of the charge carriers[3,4]. Third, the thermal conductivity can be reduced by adding a number of interfaces and phonon scattering centres in a nanowire, nanotube, superlattice, alloy or composite. To date, the majority of research has focused on increasing $\mu/\kappa$. Recently, Heremans et al.[5] emphasized the importance of the factor, $S^2 n$ where $n$ is a carrier density, on increasing ZT.

Recently, SnSe has attracted a great deal of attention for its excellent thermoelectric properties. Bulk SnSe is a well-known p-type semiconductor with an indirect band gap energy of $E_g = 0.829\,\mathrm{eV}$ at 300 K with an orthorhombic $Pnma$ phase ($a = 11.49\,\text{Å}$, $b = 4.44\,\text{Å}$, $c = 4.14\,\text{Å}$) and a direct band gap of $E_g = 0.464\,\mathrm{eV}$ with a $Cmcm$ structure phase at high temperatures (750–800 K)[6,7]. It exhibits a two-dimensional (2D) layered structure with strong tin–selenium (Sn–Se) covalent bonding along the $b$–$c$ plane and a weak Van der Waals force along the $a$ axis, which gives the material strong anisotropic transport properties[8]. Recently, Zhao et al.[9] reported that bulk SnSe is a very good p-type thermoelectric material due to its low thermal conductivity at high temperature; ZT values along the $b$ and $c$ axes are up to 2.6 and 2.3 at 923 K, respectively. More recently, two first-principle calculations predicted that both n- and p-type SnSe have high ZT values above 2. Interestingly, n-type SnSe is expected to show better thermoelectric performances than p-type SnSe[10,11]. In Na-doped p-type SnSe, Zhao et al.[12] observed a ZT value ranging from 0.7 to 2.0 over the temperature range of 300–773 K. There are a few reports on n-type doped SnSe, which are polycrystalline with ZT values of below 1; ZT = 1 in iodine-doped and 0.7 in BiCl$_3$-doped SnSe[13,14].

Here, we report that n-type SnSe single crystals were successfully synthesized by doping bismuth (Bi). Furthermore, we were able to control n-type carrier concentration by varying the Bi-doping content. The undoped SnSe is a p-type with a carrier concentration of $5.2 \times 10^{18}\,\mathrm{cm}^{-3}$ at 773 K. With various Bi-doping contents, we achieved n-type SnSe single crystals with the carrier densities of $-8.7 \times 10^{18}$, $-1.7 \times 10^{19}$ and $-2.1 \times 10^{19}\,\mathrm{cm}^{-3}$ at 773 K. Then, we obtained an excellent ZT value of 2.2 along $b$ axis at 733 K in the n-type SnSe sample with a carrier density of $-2.1 \times 10^{19}\,\mathrm{cm}^{-3}$ at 773 K.

## Results

**Doping Bi into SnSe.** SnSe has a layered structure so that the single crystal is easily cleaved along $a$ axis in the orthorhombic

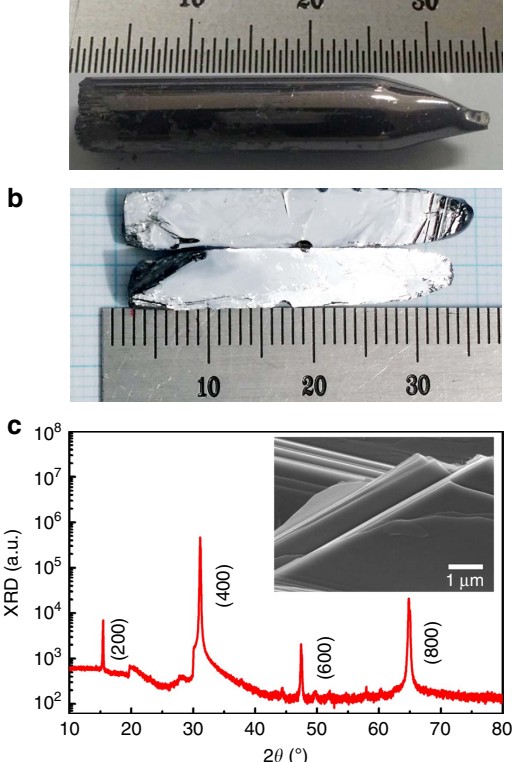

**Figure 1 | Sample photos and structural characterization.** (**a**) The photos of SnSe single crystal, (**b**) cleaved sample, (**c**) XRD pattern of SnSe single crystals. (*h00*) diffraction peaks indicate that the crystallographic $a$ axis is perpendicular to the cleaved plane of the single crystal. The inset is FE-SEM image. Note that all samples, including undoped and Bi-doped samples, show identical XRD pattern.

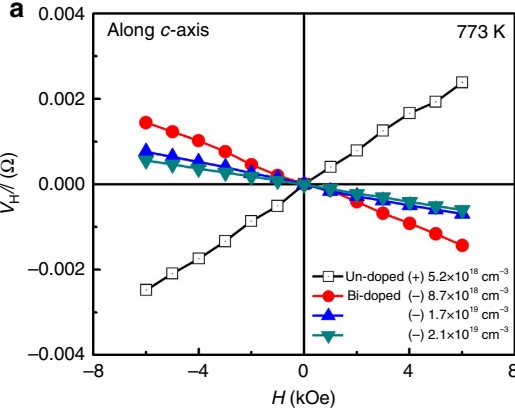

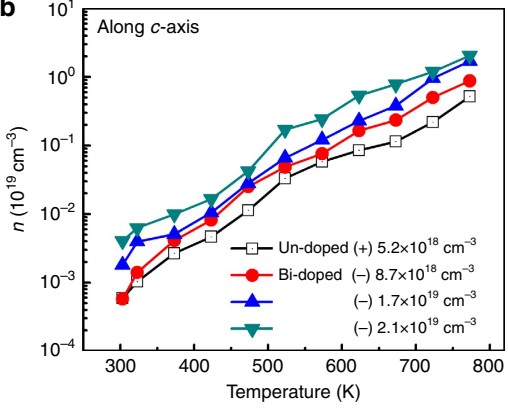

**Figure 2 | Hall transport measurements and carrier concentrations.** (**a**) $V_H/I$ versus magnetic field curves of undoped and different Bi contents—doped SnSe single crystals at 773 K. (**b**) Temperature-dependent carrier concentration of undoped and Bi-doped SnSe single crystals. The carrier concentration was determined from the linear fitting of $V_H/I$ versus magnetic field slope. Carrier concentration values at 773 K were used to distinguish samples such as ($+$) $5.2 \times 10^{18}\,\mathrm{cm}^{-3}$ for undoped SnSe, ($-$) $8.7 \times 10^{18}\,\mathrm{cm}^{-3}$ for 2% Bi, ($-$) $1.7 \times 10^{19}\,\mathrm{cm}^{-3}$ for 4% Bi and ($-$) $2.1 \times 10^{19}\,\mathrm{cm}^{-3}$ for 6% Bi at 773 K.

structure, as shown in Fig. 1a,b. The crystal structure of the cleaved Bi-doped SnSe single crystal was confirmed by the X-ray diffraction (XRD) patterns. The XRD pattern indicated an orthorhombic structure with the space group $Pnma$ of SnSe. As shown in Fig. 1c, we observed only ($h00$) diffraction peaks, indicating that the crystallographic $a$ axis is perpendicular to the cleaved plane of the single crystal. The lattice constant of the $a$ axis was determined to be 11.483 Å, comparable to the literature. Field emission scanning electron microscope (FE-SEM) images (as shown in the inset of Fig. 1c) exhibited a lamellar micro-structure with an average thickness of a few µm, which resulted from the stacking of exfoliated nano-sheets along the $a$ axis. Energy dispersive spectrometer measurements indicated the stoichiometry of Sn and Se to be 1:1, and thus a formation of orthorhombic SnSe is confirmed.

Hall resistances as a function of magnetic field of the undoped and Bi-doped SnSe single crystals at 773 K are shown in Fig. 2a. From the slope of $V_H/I$ versus the magnetic field, the carrier concentrations of samples have been calculated using the formula, $\frac{V_H}{I} = \frac{1}{ned}H$, where $V_H$ is the Hall voltage, $I$ is the current, $n$ is the number of carriers, $e$ is the electrical charge, $H$ is the magnetic field and $d$ is the sample thickness. Hall resistance indicates that the undoped SnSe exhibits a positive slope for the $V_H/I$ versus $H$ curve, indicating a p-type characteristic with a carrier concentration of $5.2 \times 10^{18}$ cm$^{-3}$ at 773 K. On the other hand, Bi-doped SnSe exhibits a negative slope, indicating successful n-type doping with various carrier densities of $-8.7 \times 10^{18}$, $-1.7 \times 10^{19}$ and $-2.1 \times 10^{19}$ cm$^{-3}$ at 773 K. We used these carrier concentrations to distinguish samples. We measured the temperature dependence of electron carrier concentrations for Bi-doped SnSe from 300 to 773 K. Electron carrier concentrations of all Bi-doped

samples exponentially increased with temperature as shown in Fig. 2b, which is typical semiconducting behaviour. Electron carrier concentrations at 300 K are $-5.7 \times 10^{15}$, $-1.8 \times 10^{16}$ and $-2.5 \times 10^{16}$ cm$^{-3}$ for three different Bi-doping contents.

**Thermoelectric transport.** Temperature-dependent Seebeck coefficients for undoped and Bi-doped SnSe single crystals along $c$ axis are shown in Fig. 3a. The undoped sample is p-type with a Seebeck coefficient of 500 µV K$^{-1}$ at 300 K, while all Bi-doped samples are n-type, with coefficients in the range of $-520$ to $-730$ µV K$^{-1}$ at 300 K. The Seebeck coefficient in the undoped SnSe increases with temperature until 650 K and then slowly decreases, while the magnitude of Seebeck coefficient in Bi-doped single crystals decreases with temperature. Thus, our measurements of Hall effects and Seebeck coefficients consistently indicate that Bi doping to SnSe induces electron carriers.

Temperature-dependent electrical conductivity ($\sigma$) of the undoped and Bi-doped SnSe along $c$ axis was shown in Fig. 3b. In the undoped sample, the electrical conductivity decreases with temperature below 600 K, while it increases in all n-type samples from room temperature to high temperatures. Increased electron carrier concentration due to Bi doping is the cause of high electrical conductivity in n-type SnSe. The temperature dependence of the thermoelectric power factor (PF) is shown in Fig. 3c. Small PF values were obtained in p-type SnSe and in lightly Bi-doped sample due to the low electrical conductivity. In relatively highly Bi-doped samples, we observed an increase in PF with temperature up to 570–670 K. The maximum PFs are 11.25 µW cm$^{-1}$K$^{-2}$ at 573 K and 10.36 µW cm$^{-1}$ K$^{-2}$ at 673 K in the samples with n-type carrier densities of $1.7 \times 10^{19}$ and $2.1 \times 10^{19}$ cm$^{-3}$ at 773 K, respectively.

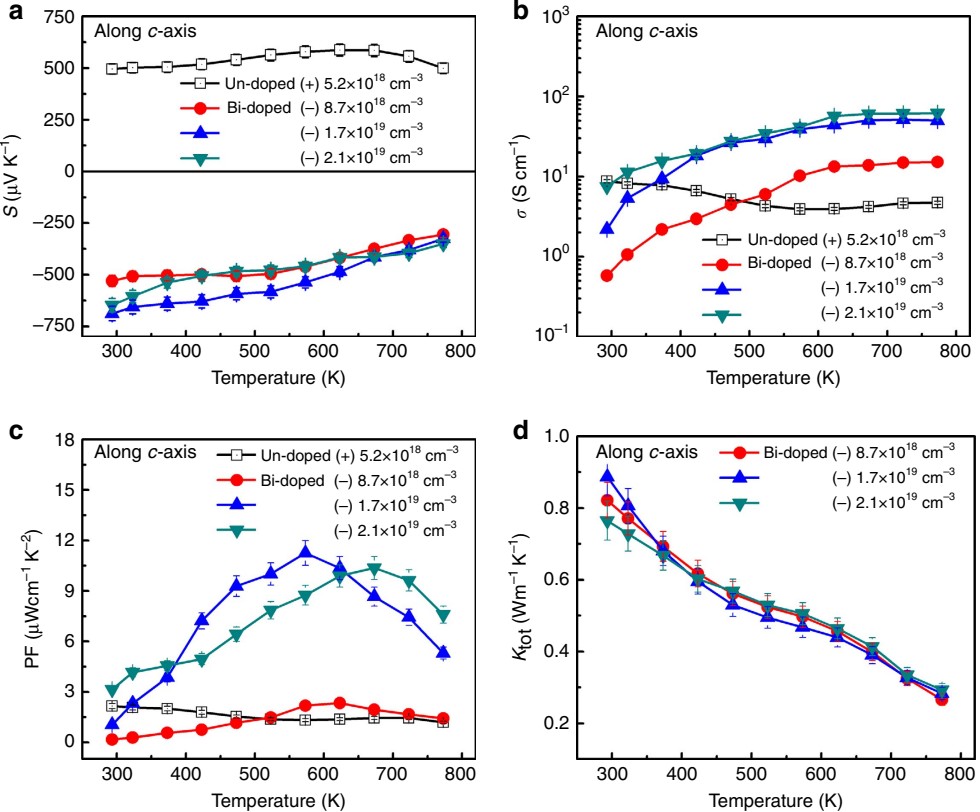

**Figure 3 | Thermoelectric properties as a function of temperature. (a)** Seebeck coefficients, **(b)** electrical conductivities and **(c)** thermoelectric PFs of undoped and different Bi contents—doped SnSe single crystals, **(d)** thermal conductivities of different Bi contents—doped SnSe single crystals. Bi-doped samples become n-type, in agreement with the Hall data. Error bars are described in Supplementary Note 1.

To determine the ZT value, the total thermal conductivity ($\kappa_{tot}$) of Bi-doped SnSe single crystals was measured from 300 to 773 K as shown in Fig. 3d. The results show that the values of $\kappa_{tot}$ for three different Bi-doping contents are similar. It should be mentioned that in semiconductors, the lattice phonons dominantly contribute to $\kappa_{tot}$. Therefore, in our n-type samples, the contribution of electron carriers on $\kappa_{tot}$ is negligible. Thermal conductivity values of n-type SnSe single crystals are comparable with the reported thermal conductivities of p-type SnSe, 0.23, 0.33 and 0.3 Wm$^{-1}$ K$^{-1}$ along the $a$, $b$ and $c$ axes at approximately 773 K, respectively[9].

ZT values along $c$ axis as a function of temperature are shown in Fig. 4. Small ZT values were observed in undoped and light Bi-doped samples. However, we found that the ZT value can be dramatically enhanced with a high carrier concentration. The maximum ZT value along $c$ axis is 2.1 at 733 K in the sample with carrier densities of $2.1 \times 10^{19}$ cm$^{-3}$, and a sample with a density of $1.7 \times 10^{19}$ cm$^{-3}$ shows a ZT value of 1.6 at 733 K.

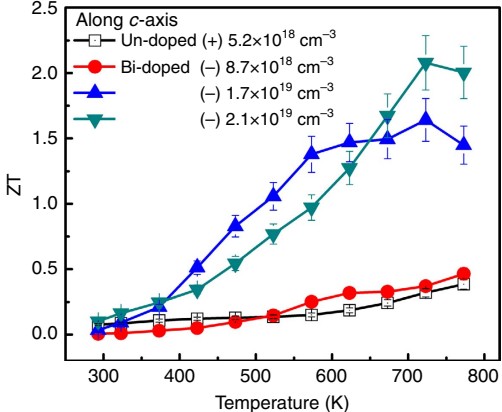

**Figure 4 | Thermoelectric figure of merit (ZT) along $c$ axis.** Small ZT values were observed in undoped and light Bi-doped samples. ZT value can be dramatically enhanced with a high carrier concentration. The maximum ZT value along $c$ axis is 2.1 at 733 K in the sample with carrier densities of $2.1 \times 10^{19}$ cm$^{-3}$. Error bars are described in Supplementary Note 1.

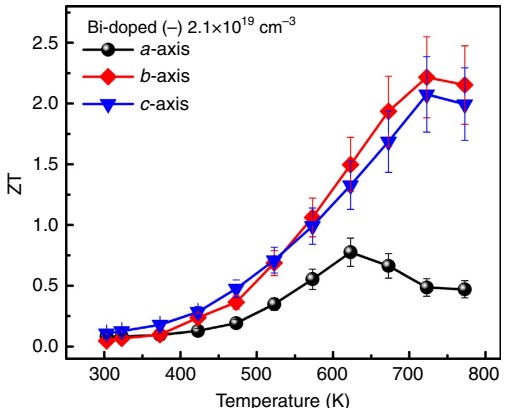

**Figure 6 | Anisotropic thermoelectric figure of merit (ZT).** Thermoelectric figure of merit (ZT) of Bi-doped SnSe sample with carrier concentration of $2.1 \times 10^{19}$ cm$^{-3}$ at 773 K along $a$–$c$ crystal axes. Maximum ZT values are 2.2, 2.1 and 0.8 along $a$–$c$ axes, respectively. Error bars are described in Supplementary Note 1.

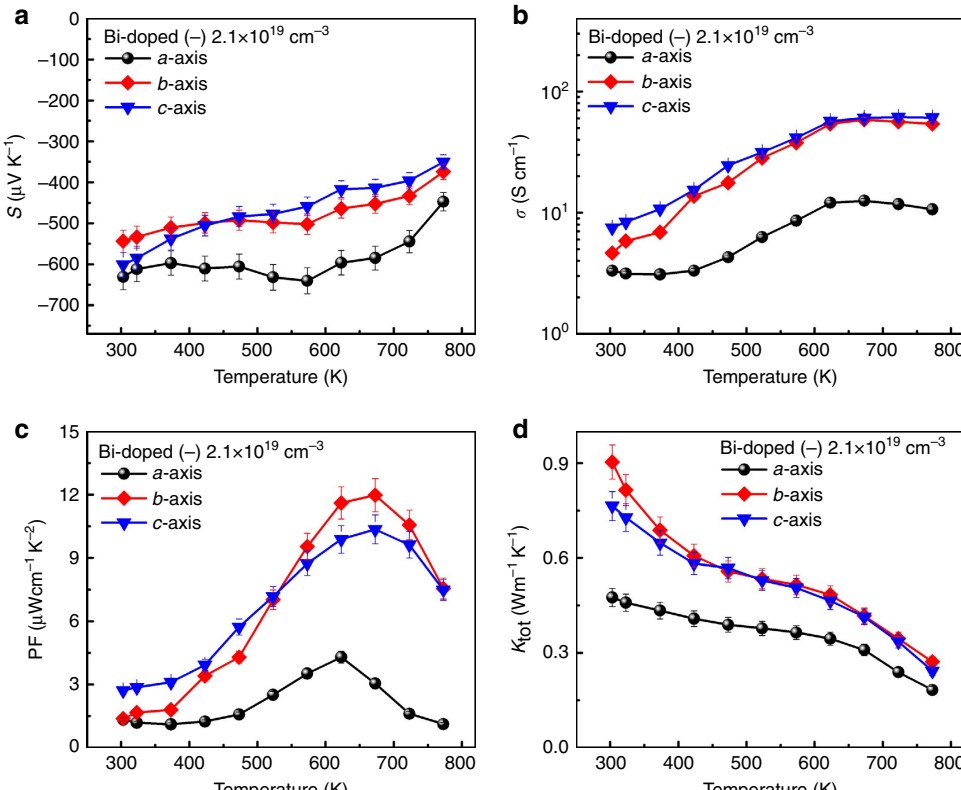

**Figure 5 | Anisotropic thermoelectric properties.** Temperature dependent (**a**) Seebeck coefficients, (**b**) electrical conductivities, (**c**) thermoelectric PFs and (**d**) thermal conductivities of Bi-doped SnSe sample with carrier concentration of $2.1 \times 10^{19}$ cm$^{-3}$ at 773 K along $a$–$c$ crystal axes. Error bars are described in Supplementary Note 1.

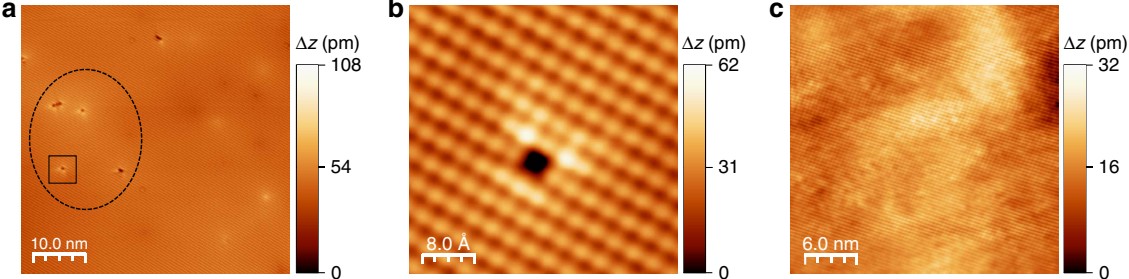

**Figure 7 | STM topographic images of undoped and Bi-doped SnSe.** (**a**) STM topographic image on the *b–c* plane of undoped SnSe (p-type). The dotted ellipse indicates Sn vacancies. (**b**) High resolution STM image taken from one of the Sn vacancies which is marked with a box in **a**. (**c**) STM topographic image on the *b–c* plane of Bi-doped SnSe (n-type). The carrier concentration for undoped and Bi-doped SnSe is $5.2 \times 10^{18}\,cm^{-3}$ at 773 K and $-8.7 \times 10^{18}\,cm^{-3}$ at 773 K, respectively. STM scanning condition: (**a,b**) sample bias $(V_b) = -1.5\,V$, tunneling current $(I_t) = 30\,pA$, (**c**) $V_b = -2.0\,V$, $I_t = 18\,pA$. The height scale $(\Delta z)$ is given by the colour bar on the right of the STM images.

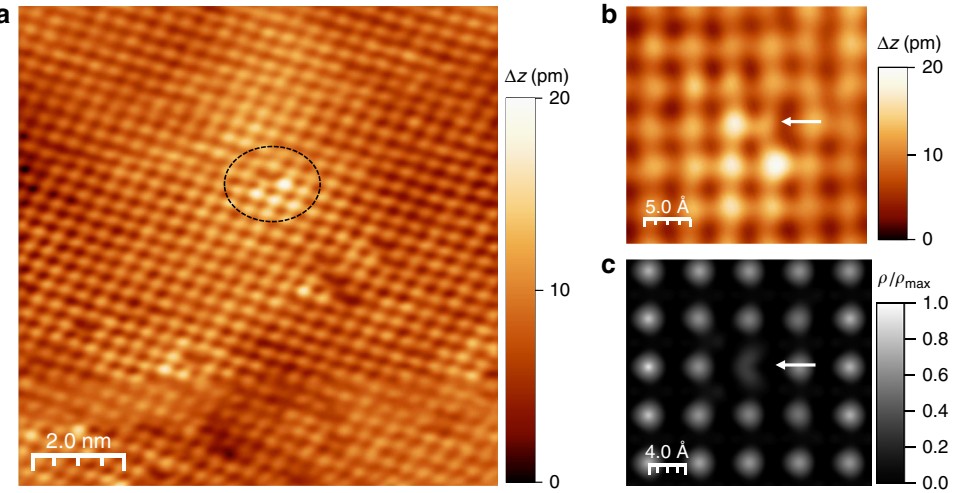

**Figure 8 | Topographic and simulated STM images of Bi-doped SnSe.** (**a**) STM topographic image of Bi-doped SnSe. The dotted ellipses indicate Bi dopants. (**b**) High resolution image of a Bi dopant taken from one of the circled areas in **a**. The height scale $(\Delta z)$ is given by the colour bar on the right of the STM images. (**c**) Simulated STM image of SnSe supercell with Bi dopant at Sn site. $\rho$ in the colour bar represents charge density $(e\,Å^{-3})$. The white arrows in **b,c** indicate the substitutional Bi atoms occupying Sn site. All STM images were taken with $V_b = -1.8\,V$ and $I_t = 30\,pA$.

**Anisotropic thermoelectric transport**. The anisotropic thermo-electric properties along *a*, *b* and *c* axes of Bi-doped SnSe with carrier concentration of $2.1 \times 10^{19}\,cm^{-3}$ were shown in Fig. 5. Anisotropic thermoelectric properties were clearly exhibited. Indeed, electrical conductivity and thermal conductivity along *b* and *c* axes are much larger than those along *a* axis, while Seebeck coefficient along *a* axis is slightly larger. So we observed maximum PF along *b* and *c* axes are 11.98 and $10.36\,\mu Wcm^{-1}\,K^{-2}$, while that along *a* axis is $4.29\,\mu Wcm^{-1}\,K^{-2}$. Temperature-dependent ZT along *a*, *b* and *c* axes are shown in Fig. 6. Highest ZT value of 2.2 was observed along *b* axis at about 733 K, while those along *c* and *a* axes are 2.1 and 0.8, respectively. This result is comparable to ZT = 2.6 at 923 K for the reported p-type SnSe[9], indicating that both n- and p-type SnSe exhibit excellent thermoelectric properties, as predicted by density functional theory (DFT) calculations[10,11]. The anisotropic trend is similar with p-type SnSe. However, the optimum temperature is 773 K in Bi-doped SnSe, whereas 923 K in p-type SnSe.

**STM microscopic study**. To clarify that Bi is substituted at the Sn site of SnSe, we carried out microscopic study on undoped (p-type) and Bi-doped (n-type) SnSe using low temperature scanning tunnelling microscope (STM) operated at 79 K under ultra-high vacuum environment $(<10^{-10}\,Torr)$. All samples were cleaved *in situ* to obtain clean *b–c* planes of SnSe. Figure 7a

exhibits a typical topographic image for undoped SnSe, where Sn vacancies (marked by a black circle) are widely distributed, indicating that Sn vacancies are dominant native defects in SnSe. The high resolution image (Fig. 7b) shows bright intensity at the Sn site, but dark intensity at the Se site with a rectangular unit cell. It is reported that this contrast difference is attributed to the upward buckling of Sn atoms and the dominant contribution of Sn 5p states[15]. On the contrary, as shown in Fig. 7c Sn vacancies were not observed in Bi-doped SnSe.

Figure 8a shows an atomic resolution image for Bi-doped SnSe surfaces. Instead of dominant Sn vacancy in undoped SnSe, new defects were found as indicated by a circle. More clearly, we found very distinctive topographic feature in Fig. 8b indicated by a white arrow, which never been observed in undoped SnSe. We propose that this unique feature is originated by substitutional Bi atoms occupying Sn site. To confirm the existence substitutional Bi atoms, we simulate the STM image for a $5 \times 5 \times 1$ SnSe supercell consisting of 100 Sn and 100 Se atoms using the DFT calculations. Single Sn atom is removed from the supercell, and then the Sn vacancy is occupied by single Bi atom. As shown in Fig. 8c, the unique shape is well reproduced in STM simulation. Therefore, we confirmed that Bi atoms are indeed occupying the Sn sites of SnSe. The success of n-type doping can be attributed to the highly similar atomic sizes of Bi (1.43 Å) and Sn (1.45 Å), which readily allow for the substitution of dopant atoms.

## Discussion

The synthesis of n-type SnSe with a ZT value of 2.2 have been achieved by Bi doping in a SnSe single crystal, and the electrical conductivity increased with Bi-doping concentration. We achieved the maximum ZT value of 2.2 along $b$ axis at 733 K in the n-type SnSe single crystal with a carrier density of $2.1 \times 10^{19}\,\mathrm{cm}^{-3}$, and this high ZT value is comparable to that of previously reported p-type SnSe. The excellent thermoelectric properties of n-type SnSe can be attributed to that the electrical conductivity is enhanced via Bi doping while maintaining low thermal conductivity. Superior thermoelectric characteristics for both n- and p-types in a one-material system are a unique advantage of SnSe. Further enhancement of ZT in n-type SnSe may be realized by optimizing doping concentration with heavier Bi doping, by changing dopants, with resonant doping, etc. For realistic device applications, it is also desirable to reduce the optimum temperature in both n- and p-type SnSe. One way might be to synthesize compound semiconductors with other materials similar in crystal structure, such as black phosphorous, GeSe and SnS.

## Methods

**Sample preparation.** The undoped and Bi-doped SnSe single crystals were grown using the temperature gradient growth method, as described elsewhere[16] We choose Bi as an n-type dopant because its atomic size (1.43 Å) is very similar to that of Sn (1.45 Å). High purity (99.999%) Sn, Se and Bi powders were used for the sample growth. First, Sn and Se powders with a total weight of 20 g were loaded into thick wall quartz ampoules. In addition, we added Bi into the ampoule for doping, using amounts that are 2, 4 and 6% Sn. Then, the ampoules were evacuated above $10^{-4}$ Torr and sealed. Another quartz tube was sealed to protect the sample and ampoule when the ampoule breaks during heating due to high vapour pressure of Se and/or cooling, due to the different thermal expansion coefficient between SnSe and quartz ampoule. The powders in ampoule were mixed and loaded into a vertical furnace. The temperature was slowly increased ($10\,^{\circ}\mathrm{C}\,\mathrm{h}^{-1}$) from room temperature to 930 °C and maintained at this temperature for 10 h. Finally, the temperature was slowly cooled ($1\,^{\circ}\mathrm{C}\,\mathrm{h}^{-1}$) from 930 to 700 °C and then rapidly ($20\,^{\circ}\mathrm{C}\,\mathrm{h}^{-1}$) from 700 °C to room temperature.

**Sample characterization.** XRD and FE-SEM were used to investigate the crystal structure. The composition was determined by an energy dispersive spectrometer measurement. Electrical resistivity and Seebeck coefficients were measured under an Ar atmosphere from room temperature to 783 K. The laser flash diffusivity method (model: LFA-457, NETZSCH, Germany) was used to determine thermal diffusivity from room temperature to 773 K. Thermal conductivity was calculated with the formula $\kappa_{\mathrm{tot}} = DC_{\mathrm{P}}\rho$, where $D$ is the thermal diffusivity, $C_{\mathrm{P}}$ is the heat capacity and $\rho$ is the density. For STM study, a single-crystal SnSe sample was transferred into the home-built low temperature STM system and cleaved *in situ* to obtain clean surfaces[17]. Tungsten tips were prepared by electrochemical etching and cleaned with electron beam heating for STM measurements. The DFT calculations were performed using the generalized gradient approximation and the projector-augmented wave method with a plane-wave basis as implemented in the Vienna ab initio simulation package code.

**Data availability.** The data that support the findings of this study are available from the corresponding author on reasonable request.

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

## Acknowledgements

This work was supported by the National Research Foundation of Korea [NRF-2009-0093818 and NRF-2014R1A4A1071686], by the Korea Evaluation Institute of Industrial Technology (KEIT) funded by the Ministry of Trade, Industry and Energy (MOTIE) (Project No. 10050296, Large scale (Over 8″) synthesis and evaluation technology of 2-dimensional chalcogenides for next generation electronic devices) and by a grant from the Energy Efficiency & Resources program of the Korea Institute of Energy Technology Evaluation and Planning (KETEP) funded by the Korean Ministry of Knowledge Economy (20132020000110).

## Author contributions

A.T.D., V.Q.N. and V.T.D. synthesized the high quality SnSe and Bi-doped SnSe single crystals and performed the transport measurements. S.K., J.K.L., S.D.P. and J.E.L. measured thermal conductivity, S.C. initiated the study and edited the manuscript, J.K. edited the manuscript, G.D. and J.K. performed the STM experiments and analysed the STM data, T.M. and J.L. conducted the first-principles calculation, A.T.D. wrote the paper with discussion and comments from all the authors. J.Y.S. and S.C. supervised the project.

## Additional information

**Competing financial interests:** The authors declare no competing financial interests.

**Publisher's note**: 

