## [Peer Review File · Nature Communications]

Reviewer #1 (Remarks to the Author)

Review for NCOMMS-16-09669:

To the Authors

A. Summary of the key results:

Doping SnSe single crystals with Bi produces a $ZT = 2.1$ at 773 K with carrier density of $2.5 \times 10^{16} \text{ cm}^{-3}$

B. Originality and interest: if not novel, please give references:

Compliments the previous work of Zhao et al. Nature 508, 2014 for device applications. Result important to those involved in thermoelectric device applications.

C. Data & methodology: validity of approach, quality of data, quality of presentation:

Data is clear however the manner in which you reference samples is ambiguous. See additional comments below.

D. Appropriate use of statistics and treatment of uncertainties:

Analysis proper.

E. Conclusions: robustness, validity, reliability:

Conclusions seem sound.

F. Suggested improvements: experiments, data for possible revision:

See detailed comments below. The manner in which samples are referenced should be changed.

No additional experiments need to be undertaken to support conclusion.

G. References: appropriate credit to previous work?

Appropriate credit is given to previous authors.

H. Clarity and context: lucidity of abstract/summary, appropriateness of abstract, introduction and conclusions

The MS is clear and appropriate. Changes suggested below should be implemented before editor considers publication.

+++

To the Author:

Overall this is a fine paper and an important new result for the thermoelectric field. This paper will encourage additional doping into the single crystal SnSe system. This n-type material pairs well for device applications with the p-type thermoelectric SnSe with a slightly higher ZT. This paper/research follows directly upon the research of Zhao et al. Nature 508, 2014 for p-type single crystals of SnSe ($ZT=2.6$) and compliments it.

The manner in which the samples are referenced in the paper by carrier concentration rather than % Bi doping is a necessary correction to the text and figures. My further comments/corrections are:

MS currently states: Title of MS

Achievement of $ZT=2.1$ in Bi-doped n-type SnSe single crystals

Referee suggested correction: Title of MS

Achieving $ZT=2.1$ with Bi-doped n-type SnSe single crystals

+++

MS reads on Page 1, Line 15

time we report that n-type SnSe single crystal was successfully synthesized by

Referee Correction Page 1, Line 15

time we report that an n-type SnSe single crystal was successfully synthesized by

or alternatively

time we report that n-type SnSe single crystals were successfully synthesized by

+++

MS reads on Line 112

which is 0.29 W/mK along c at 773 K.

Referee Correction on Line 112

which is 0.29 W/m-K along c at 773 K.

+++

MS reads on Line 113

which are 0.23, 0.33, and 0.3 W/mK

Referee Correction on Line 113

which are 0.23, 0.33, and 0.3 W/m-K

+++

MS reads on Line 151-152

Additionally, we added Bi into the ampoule for doping, using amounts that are 2%, 4%, and 6% Sn.

Referee Comment on Line 151-152

This method of reporting chemical doping is ambiguous. Are the attempted formula weights of the samples: $\text{Bi}_{0.02}\text{Sn}_{0.98}\text{Se}_{1.00}$, $\text{Bi}_{0.04}\text{Sn}_{0.96}\text{Se}_{1.00}$, and $\text{Bi}_{0.06}\text{Sn}_{0.94}\text{Se}_{1.00}$? If this is correct then, it should be stated properly by the authors. The figures that reference each of these doping concentrations are not clearly indicated. While the authors indicate the sample carrier concentrations to distinguish the sample performance, this phenomenon is driven by the % Bi doping. The data should reference % Bi or formula weight to distinguish the samples. The data labels in Figs. 2, 3, 4, and 5 should reflect % Bi or formula weight concentration for the samples.

+++

MS reads on Line 183

electronic transitions in SnS and SnSe semiconductors.

Referee Correction on Line 183

electronic transitions in SnS and SnSe semiconductors.

+++

MS reads on Line 200

J. Magn. Magn. Mater. 304, 164-166 (2006).

Referee Correction on Line 200

J. Magn. Magn. Mater. 304, 1, e164-e166 (2006).

+++

MS reads on Line 203-205

FIG 1 (a) XRD pattern and (b) FE-SEM image of SnSe single crystals. Note that all samples, including un-doped and Bi-doped samples, show identical XRD pattern

Referee Comment/Correction on Line 203-205

There is no reference scale on for the insert picture in Fig. 1a; one should be provided by the authors. The reader cannot determine if these crystals are 0.1 mm, 1 mm or 10 mm in length.

+++

MS reads on Line 206

FIG 2 VH/I vs magnetic field linear curves

Referee Correction on Line 206

FIG 2 VH/I vs. magnetic field linear curves

+++

MS reads on Line 302

SnSe single crystals and transport measurement. S. K, and J. K. L. measured

Referee Correction on Line 302

SnSe single crystals and performed the transport measurements. S. K, and J. K. L. measured

+++

MS reads on Line 304

the manuscript, A.T.D wrote the paper with discussed and commented from all the

Referee Correction on Line 304

the manuscript, A.T.D wrote the paper with discussion and comments from all the

Reviewer #2 (Remarks to the Author)

The authors reported the synthesis and characterization of n-type SnSe single crystal. While the result will be interesting to the thermoelectric community, it should not claim "no success result on n-type doping of SnSe has been reported by the community thus far" (see APL 108, 083902,

arXiv: 1601.00753v2, and 10.1002/aenm.201500360 as a few examples). The synthesis of Bi-doped single crystal SnSe does allow the measurements of properties along crystalline axial directions. The results appear reasonable. However, it is doubtful that the manuscript represents any advance in understanding thermoelectricity or SnSe or influences thinking in the field. As a result, I would recommend the manuscript being published in a more applied journal. Some other comments I have with the manuscript.

1) Error bars are obviously missing in many figures. Even in the cases where the error bars are hard to quantify, an estimation is still better than leaving them out.

2) The authors believe that Bi occupies the Sn sites. It is likely but not supported.

3) Thermal conductivity of different doped samples should be shown. Even though the electronic contribution is small, the effect of doping on phonons at a few atom% can be quite large.

4) The figures in supplemental materials should be improved to similar quality as the main text.

Reviewer #3 (Remarks to the Author)

The authors report on unrepresented, and extremely high values of ZT reached in n-doped SnSe. This topic could be of large interest for the readership of Nature Communications. However, the paper contains some very unexpected results that raise some important questions that have to be addressed by the authors. Thus, I suggest rejection, yet encourage the authors to resubmit if they address the mandatory points mentioned below:

- The authors write that « the single crystal is easily cleaved along a-axis [...] as shown on Figure 1a ». Actually, Figure 1a shows an ingot with a very rough and irregular cleavage, what does not agree with the statement of the authors. Besides, it would be interesting to know the opinion of the authors regarding the orientation of the crystal during the growth.
- The carrier concentration in the un-doped sample is found to be $6 \times 10^{15} \text{ cm}^{-3}$, what is about 100 times less than what have been measured by Zhao et al. [9]. The authors should comment on this major difference, taking into account that DFT calculations have shown that the vacancy of tin is very probable (very low enthalpy of formation), and does induce a large density of charge carriers (see Bera et al. Phys.Chem.Chem.Phys. 16, 19894 (2014)).
- As expected (because of the low enthalpy of formation of the vacancy of tin, which is a "killer" p-doping defect) the doping level reached by the authors is very small (only a few 10^{16} cm^{-3}). In other words, the samples are barely n-doped (lower doping level than the "undoped" sample of Zhao et al. [9]), showing a carrier density only about 5x larger than the un-doped samples. This is far from being "highly doped", as claimed by the authors. This major point should be discussed in the revised manuscript, especially taking into account the prior art, both experimental and DFT based.
- The authors claim that DFT calculation have forecasted high TE properties for n-doped SnSe. This is true, but only for very high charge carrier concentrations (above 10^{19} cm^{-3}). The regime in which the authors operate is very far from the necessary concentrations, and the large ZT values claimed by the authors should not occur in such low concentration of carriers. The author should comment, and explain this very strong difference.
- In order to convince the reader about the very high value of ZT claimed, the authors should prove that the ZT values are strongly anisotropic, and measure the TE properties (thermal and electrical conductivity as well as Seebeck) along the three different axis of the single crystal.

Reviewer #1:

We thank the reviewer for recognizing the significance of our work. We appreciate very much the reviewer's constructive comments and suggestions, which are very helpful for us to sharpen the messages better in the revised manuscript. Some of your comments we would like to detail explain as below:

- 1) The referee commented that *“This method of reporting chemical doping is ambiguous. Are the attempted formula weights of the samples: $Bi_{0.02}Sn_{0.98}Se_{1.00}$, $Bi_{0.04}Sn_{0.96}Se_{1.00}$, and $Bi_{0.06}Sn_{0.94}Se_{1.00}$? If this is correct then, it should be stated properly by the authors. The figures that reference each of these doping concentrations are not clearly indicated. While the authors indicate the sample carrier concentrations to distinguish the sample performance, this phenomenon is driven by the % Bi doping. The data should reference % Bi or formula weight to distinguish the samples. The data labels in Figs. 2, 3, 4, and 5 should reflect % Bi or formula weight concentration for the samples.”*

Answer:

For samples preparation, we added Bi into the ampoule for doping, using amounts that are corresponding to 2%, 4%, and 6% Sn. However, not all amounts of Bi were substituted into Sn sites in SnSe crystal, implying that some Bi still remained at the top of sample ingots. This is the reason why we could not distinguish samples by initial Bi content. Instead, we identified our samples with carrier concentrations such as $8.7 \times 10^{18} \text{ cm}^{-3}$ for 2% Bi, $1.7 \times 10^{19} \text{ cm}^{-3}$ for 4% Bi, and $2.1 \times 10^{19} \text{ cm}^{-3}$ for 6% Bi at 773 K and such as $5.7 \times 10^{15} \text{ cm}^{-3}$ for 2% Bi, $1.8 \times 10^{16} \text{ cm}^{-3}$ for 4% Bi, and $2.5 \times 10^{16} \text{ cm}^{-3}$ for 6% Bi at 300 K.

- 2) The referee commented that *“There is no reference scale on for the insert picture in Fig. 1a; one should be provided by the authors. The reader cannot determine if these crystals are 0.1 mm, 1 mm or 10 mm in length.”*

Answer:

We replaced the sample photo in Fig. 1(a). In the new photo, a ruler is shown to indicate the size of Bi-doped SnSe single crystal.

- 3) The referee suggested *the title change from “Achievement of $ZT=2.1$ in Bi-doped n-type SnSe single crystals” to “Achieving $ZT=2.1$ with Bi-doped n-type SnSe single crystals”.*

Answer:

The title is changed in the revised manuscript as the referee suggested.

Additionally, we measured the ZT values along all three crystal directions in the revised manuscript; in fact, we achieved the ZT value of 2.2 along b axis at 733 K (we obtained $ZT = 2.1$ along c-axis in the previous manuscript). Thus, we updated this ZT value of 2.2 in the title of revised manuscript.

Reviewer #2:

We appreciate very much his/her constructive comments and suggestions. Below we address his/her specific comments in detail.

- 1) The referee commented that *“While the result will be interesting to the thermoelectric community, it should not claim “no success result on n-type doping of SnSe has been reported by the community thus far” (see APL 108, 083902, arXiv:1601.00753v2, and 10.1002/aenm.201500360 as a few examples).”*

Answer:

We again appreciate referee’s comment on this point. As a matter of fact, there are some reports on n-type SnSe. However, all of them reported about n-type doped polycrystalline SnSe, and they were not successful in achieving ZT values comparable to $ZT = 2.6$ of p-type SnSe: $ZT = 1$ in Iodine doped SnSe¹ and $ZT = 0.7$ in BiCl₃ doped SnSe.² In order to address the referee’s comment, we revised the sentence on **line 61, page 3** as following;

FROM: “However, no successful result on n-type doping of SnSe has been reported to the community thus far. Here, we report that n-type SnSe single crystals were successfully synthesized by doping Bi into Sn sites for the first time.”

TO: “However, no successful experiment results on n-type SnSe with ZT values comparable to $ZT = 2.6$ of p-type SnSe have been reported to the community thus far. Here, we report that n-type SnSe single crystals were successfully synthesized by doping Bi into Sn sites.”

Additionally, we removed the phrase “for the first time” in **line 18, page 1**.

We added a below sentence in **line 56, page 3**.

“There are a few reports on doped n-type SnSe, which are polycrystalline with ZT values of below 1; $ZT = 1$ in Iodine-doped and 0.7 in BiCl₃-doped SnSe.^{1,2}”

- 2) The referee commented that “Error bars are obviously missing in many figures. Even in the cases where the error bars are hard to quantify, estimation is still better than leaving them out.”

Answer:

We determined the error levels of three transport properties (electrical conductivity, Seebeck coefficient, and thermal conductivity) as below.

- For electrical conductivity, $\sigma = \frac{1}{\rho} = \frac{l}{Rab}$ where ρ is resistivity, R is resistance, l is length of sample, and a and b are width and height of sample.

$$\text{Average of electrical conductivity } \langle \sigma \rangle = \frac{\langle l \rangle}{\langle a \rangle \langle b \rangle \langle R \rangle}$$

$$\text{Error of } \sigma \text{ is } \Delta \sigma = \langle \sigma \rangle \left\{ \frac{\Delta l}{\langle l \rangle} + \frac{\Delta a}{\langle a \rangle} + \frac{\Delta b}{\langle b \rangle} + \frac{\Delta R}{\langle R \rangle} \right\}$$

Sample sizes; $l = 2$ mm; $a = 2$ mm; $b = 0.3$ mm, and $R = V/I$ (Keithley model 2400 series with error less than 0.1%); $\Delta l = \Delta a = \Delta b = 0.01$ mm then

$$\Delta \sigma = \langle \sigma \rangle \left\{ \frac{0.01}{2} + \frac{0.01}{2} + \frac{0.01}{0.3} \right\} = 0.043 * \langle \sigma \rangle$$

Error for electrical conductivity measurement is 4.3%.

- For Seebeck coefficient measurements. To determine error for Seebeck coefficient, we used a Bi sample whose Seebeck coefficient is well known; i.e. we measured the Seebeck coefficients of polycrystalline Bi for 18 times as shown in below figures. We determined Seebeck coefficients from the slope of thermoelectric voltage ΔV vs. temperature gradient ΔT curve. The measured Seebeck coefficient of polycrystalline Bi is -57.7 ± 0.76 $\mu V/K$. Standard error from the linear fitting is 1.3%. Note that the Seebeck coefficient of Bi single crystal is -51.4 $\mu V/K$ along perpendicular to the three fold axis and -102.7 $\mu V/K$ along parallel to the three fold axis.³ The Seebeck coefficient of the reported polycrystalline Bi is -60 $\mu V/K$ at 300 K.⁴ The slight difference between the measured and reported values are due to the sample quality such as grain size, purity, etc.
- For thermal conductivity measurement, we used Laser Flash Apparatus LFA 457 MicroFlash measurement system with the error of $\pm 3\%$.
- Power factor (PF) is calculated by Seebeck coefficient and electrical conductivity as below:

$$PF = S^2\sigma$$

So PF error is estimated:

$$\frac{\Delta PF}{\langle PF \rangle} = \left(\frac{\Delta S}{\langle S \rangle} + \frac{\Delta S}{\langle S \rangle} + \frac{\Delta \sigma}{\langle \sigma \rangle} \right) = (0.013 + 0.013 + 0.043) = 0.069$$

- Power ZT error is estimated:

$$\frac{\Delta ZT}{\langle ZT \rangle} = \left(\frac{\Delta S}{\langle S \rangle} + \frac{\Delta S}{\langle S \rangle} + \frac{\Delta \sigma}{\langle \sigma \rangle} + \frac{\Delta \kappa}{\langle \kappa \rangle} \right) = (0.013 + 0.013 + 0.043 + 0.03) = 0.099$$

Power factor and ZT error in our measurements are 6.9% and 9.9%, respectively and we added error bars in the figures.

<Figure r1. (a) Thermoelectric voltage ΔV as a function of temperature gradient ΔT ; the linear slopes indicate the Seebeck coefficients of polycrystalline Bi. (b) Seebeck coefficient as a function of number of measurement; red line is the result of linear fitting. Fitting parameters in inset box indicate the average of Seebeck coefficients for 18 times measurements and standard error. 1.3%.>

- 3) The referee commented that *“The authors believe that Bi occupies the Sn sites. It is likely but no supported.”*

Answer:

We carried out microscopic study on un-doped (p-type) and Bi-doped (n-type) SnSe using low temperature scanning tunneling microscope (STM) operated at 79 K under ultra-high vacuum environment ($< 10^{-10}$ Torr). All samples were cleaved *in-situ* to obtain clean b-c planes of SnSe. Figure r2(a) exhibits a typical topographic image for un-doped SnSe where Sn vacancies (marked by a black circle) are widely distributed, indicating that Sn vacancies are dominant native defects in SnSe. The high resolution image (inset in Fig. r2(a)) shows bright intensity at the Sn site, but dark intensity at the Se site with a rectangular unit cell. It is reported that this contrast difference is attributed to the upward buckling of Sn atoms and the dominant contribution of Sn 5p states.⁵ On the contrary, as shown in Fig. r2(b) Sn vacancies were not observed in Bi-doped SnSe.

Figure r3(a) shows an atomic resolution image for Bi-doped SnSe surfaces. Instead of dominant Sn vacancy in un-doped SnSe, new defects were found as indicated by circles. More clearly, we found the very distinctive topographic feature in Fig. r3(b) indicated by a white arrow, which never been observed in un-doped SnSe. We propose that this unique feature is originated by substitutional Bi atoms occupying Sn site. In order to confirm the existence substitutional Bi atoms, we simulate the STM image for a $5 \times 5 \times 1$ SnSe supercell consisting of 100 Sn and 100 Se atoms using the density functional theory (DFT) calculations. Single Sn atom is removed from the supercell, and then the Sn vacancy is occupied by single Bi atom. As shown in Fig. 3r(c), the unique shape is well reproduced in the STM simulation. Therefore, we confirmed that Bi atoms are indeed occupying the Sn sites of SnSe.

We added this information in the revised manuscript.

<Figure r2. STM topographic images on the b-c plane of (a) un-doped SnSe (p-type) and (b) Bi-doped SnSe (n-type). The inset in Fig. r2.(a) shows the high resolution image of Sn vacancy (4 nm × 4 nm). The carrier concentration for un-doped and Bi-doped SnSe is $5.9 \times 10^{15} \text{ cm}^{-3}$ and $-5.7 \times 10^{15} \text{ cm}^{-3}$ at room temperature, respectively. Image condition: (a) sample bias (V_b) = - 1.5 V, tunneling current (I_t) = 30 pA, (b) V_b = - 2.0 V, I_t = 18 pA. Image condition: (a) V_b = - 1.5 V, I_t = 30 pA, (b) V_b = - 2.0 V, I_t = 18 pA.>

<Figure r3. (a) STM topographic images of Bi-doped SnSe. (b) High resolution image of Bi dopant taken from one of the circled areas in (a). (c) Simulated STM image of SnSe supercell with Bi dopant at Sn site. All STM images were taken with V_b = - 1.8 V and I_t = 30 pA.>

- 4) The referee commented that “*Thermal conductivity of different doped samples should be shown. Even though the electronic contribution is small, the effect of doping on phonons at a few atom% can be quite large.*”

Answer:

Thermal conductivities of different Bi doping content samples were measured from 300 to 800 K, following referee’s comment. Results are very similar each other, which are added in Fig. 3(d) of revised manuscript.

<Figure r4. Thermal conductivities of Bi-doped SnSe single crystals>

- 5) The referee commented that “*The figures in supplemental materials should be improved to similar quality as the main text.*”

Answer:

The quality of figures in supplemental information has been improved.

Reviewer #3:

We appreciate very much the reviewer's constructive comments and suggestions. As encouraged by the referee, we address his/her specific comments in detail below.

- 1) The referee commented that "*The authors write that « the single crystal is easily cleaved along a-axis [...] as shown on Figure 1a ». Actually, Figure 1a shows an ingot with a very rough and irregular cleavage, what does not agree with the statement of the authors. Besides, it would be interesting to know the opinion of the authors regarding the orientation of the crystal during the growth.*"

Answer:

We replaced the old sample photo in figure 1(a) as the referee suggested.

< Figure r5. The photos of Bi doped SnSe single crystal and cleaved sample.>

Opinion in single crystal orientation:

We observed that SnSe single crystal prefers to grow along the c-axis as shown in Fig. r5. The crystal orientation is added in Fig. 1(a) of revised manuscript. SnSe has layered orthorhombic structure with strong Sn-Se covalent bonding along b-c

plane and each layer along a-axis is weakly coupled by Van der Waals interaction. The lack of chemical bonding might be the reason why SnSe is not likely to grow along the a-axis. Along b-c plane, it is not easy to exactly know why SnSe grows along c-axis. During growth process, crystal orientation depends on many parameters such as the crystallographic orientation of the seeds and atomic bonding. In the case of SnSe, however, the bonding structure is very complicated along b and c-axes. Therefore, we can only guess that SnSe seed-crystals are preferentially oriented along the c-axis due to thermodynamic effects at the solidification point of growth process.

- 2) The referee commented that *“The carrier concentration in the un-doped sample is found to be $6 \times 10^{15} \text{ cm}^{-3}$, what is about 100 times less than what have been measured by Zhao et al. [9]. The authors should comment on this major difference, taking into account that DFT calculations have shown that the vacancy of tin is very probable (very low enthalpy of formation), and does induce a large density of charge carriers (see Bera et al. Phys.Chem.Chem.Phys. 16, 19894 (2014)).”*

Answer:

Yes, you are right. In our un-doped sample, carrier concentration is about 100 times less than report of Zhao et al. We expect that the carrier density difference between Zhao et al.'s and our values may be from the quality of single crystal (for example, Sn vacancy) due to crystal growth technique and parameters, etc. We used temperature gradient method; the ampoule temperature changes at a rate of 1°C/h . Zhao et al. used Bridgman method; the ampoule moves with the rate of 2 mm/h.

- 3) The referee commented that *“As expected (because of the low enthalpy of formation of the vacancy of tin, which is a “killer” p-doping defect) the doping level reached by the authors is very small (only a few 10^{16} cm^{-3}). In other words, the samples are barely n-doped (lower doping level than the “un-doped” sample of Zhao et al. [9]), showing a carrier density only about 5x larger than the un-doped samples. This is far of being “highly doped”, as claimed by the authors. This major point should be discussed in the revised manuscript, especially taking into account the prior art, both experimental and DFT based.*

- 4) *The authors claim that DFT calculation has forecasted high TE properties for n-doped SnSe. This is true, but only for very high charge carrier concentrations (above 10^{19} cm^{-3}). The regime in which the authors operate is very far from the necessary concentrations, and the large ZT values claimed by the authors should not occur in such low concentration of carriers. The author should comment, and explain this very strong difference.”*

Answer:

We measured the temperature dependence of electron carrier density for Bi-doped SnSe single crystals determined from Hall effect measurement. At room temperature, carrier concentrations of n-type Bi doped SnSe samples are smaller than un-doped SnSe of Zhao et al. However, with increasing temperature, the number of electron carriers in the conduction band exponentially increases with temperature, i. e., donor electrons (which were created by Bi substitution into Sn sites) are excited into the conduction band. At 773 K, the electron concentrations for three Bi-doped samples are 8.7×10^{18} , 1.7×10^{19} , and $2.1 \times 10^{19} \text{ cm}^{-3}$.

We added this data in figure 2(b) in revised manuscript. Highest ZT of 2.2 is observed in n-type SnSe single crystal with the carrier concentration of $2.1 \times 10^{19} \text{ cm}^{-3}$ at 773 K.

<Figure r7. Temperature dependent electron carrier density of Bi-doped SnSe single crystals determined from Hall effect measurement.>

We carried out microscopic study on un-doped (p-type) and Bi-doped (n-type) SnSe using low temperature scanning tunneling microscope (STM) operated at 79 K under ultra-high vacuum environment ($< 10^{-10}$ Torr). All samples were cleaved *in-situ* to obtain clean b-c planes of SnSe. Figure r8(a) exhibits a typical topographic image for un-doped SnSe where Sn vacancies (marked by a black circle) are widely distributed, indicating that Sn vacancies are dominant native defects in SnSe. The high resolution image (inset in Fig. r8(a)) shows bright intensity at the Sn site, but dark intensity at the Se site with a rectangular unit cell. It is reported that this contrast difference is attributed to the upward buckling of Sn atoms and the dominant contribution of Sn 5p states.⁵ On the contrary, as shown in Fig. r8(b) Sn vacancies were not observed in Bi-doped SnSe.

Figure r9(a) shows an atomic resolution image for Bi-doped SnSe surfaces. Instead of dominant Sn vacancy in un-doped SnSe, new defects were found as indicated by circles. More clearly, we found the very distinctive topographic feature in Fig. r9(b) indicated by a white arrow, which never been observed in un-doped SnSe. We propose that this unique feature is originated by substitutional Bi atoms occupying Sn site. In order to confirm the existence substitutional Bi atoms, we simulate the STM image for a $5 \times 5 \times 1$ SnSe supercell consisting of 100 Sn and 100 Se atoms using the density functional theory (DFT) calculations. Single Sn atom is removed from the supercell, and then the Sn vacancy is occupied by single Bi atom. As shown in Fig. 9r(c), the unique shape is well reproduced in STM simulation. Therefore, we confirmed that Bi atoms are indeed occupying the Sn sites of SnSe.

We added this information in the revised manuscript.

<Figure r8. STM topographic images on the b-c plane of (a) un-doped SnSe (p-type) and (b) Bi-doped SnSe (n-type). The inset in Fig. r2. (a) shows the high resolution image of Sn vacancy (4 nm \times 4 nm). The carrier concentration for un-doped and Bi-doped SnSe is $5.9 \times 10^{15} \text{ cm}^{-3}$ and $-5.7 \times 10^{15} \text{ cm}^{-3}$ at room temperature, respectively. Image condition: (a) sample bias (V_b) = - 1.5 V, tunneling current (I_t) = 30 pA, (b) V_b = - 2.0 V, I_t = 18 pA. Image condition: (a) V_b = - 1.5 V, I_t = 30 pA, (b) V_b = - 2.0 V, I_t = 18 pA.>

<Figure r9. (a) STM topographic images of Bi-doped SnSe. (b) High resolution image of Bi dopant taken from one of the circled areas in (a). (c) Simulated STM image of SnSe supercell with Bi dopant at Sn site. All STM images were taken with V_b = - 1.8 V and I_t = 30 pA.>

- 5) The referee commented that “*In order to convince the reader about the very high value of ZT claimed, the authors should prove that the ZT values are strongly anisotropic, and measure the TE properties (thermal and electrical conductivity as well as Seebeck) along the three different axis of the single crystal.*”

Answer:

Based on referee’s suggestion, we conducted thermoelectric measurements along all three crystal directions for the high ZT sample with carrier concentration of $2.1 \times 10^{19} \text{ cm}^{-3}$ at 773 K, as shown in figures r10 and r11 in revised manuscript. We achieved $ZT=2.2$ along b-axis, which is higher than $ZT=2.1$ along c-axis (the original manuscript’s value). The title has been changed to “Achieving $ZT=2.2$ with Bi-doped n-type SnSe single crystals”. Note that maximum ZT value along a-axis is 0.8. The anisotropic trend is similar with p-type SnSe. However, the optimum temperature is 773 K in Bi-doped SnSe, whereas 923 K in p-type SnSe.

<Figure r10. Temperature dependent (a) Seebeck coefficients, (b) electrical conductivities, (c) thermoelectric power factors, and (d) thermal conductivities of Bi-doped SnSe sample with carrier concentration of $2.1 \times 10^{19} \text{ cm}^{-3}$ at 773 K along a, b, and c-axes.>

<Figure r11. Thermoelectric figure of merit (ZT) of Bi-doped SnSe sample with carrier concentration of $2.1 \times 10^{19} \text{ cm}^{-3}$ at 773 K along a, b, and c-axes. Maximum ZT values are 2.2, 2.1, and 0.8 along b, c, and a-axes, respectively.>

References

1. Zhang, Q. *et al.* Studies on Thermoelectric Properties of n-type Polycrystalline SnSe_{1-x}S_x by Iodine Doping. *Adv. Energy Mater.* **5**, 1500360 (2015).
2. Wang, X. *et al.* Optimization of thermoelectric properties in n-type SnSe doped with BiCl₃. *Appl. Phys. Lett.* **108**, 083902 (2016).
3. Chandrasekhar, B. S. The seebeck coefficient of bismuth single crystals. *J. Phys. Chem. Solids* **11**, 268–273 (1959).
4. Ishikawa, Y., Suzuki, A., Komine, T., Shirai, H. & Hasegawa, Y. Seebeck coefficient and resistivity measurement of polycrystalline Bi in a magnetic field. *Int. Conf. Thermoelectr. ICT, Proc.* **2003-Janua**, 286–289 (2003).
5. Kim, S., Duong, A.-T., Cho, S., Rhim, S. H. & Kim, J. A microscopic study investigating the structure of SnSe surfaces. *Surf. Sci.* **651**, 5–9 (2016).

Reviewer #2 (Remarks to the Author)

The authors made major revisions to the original manuscript and addressed most of my concerns, especially the ones regarding the doping sites and the thermal conductivity of different samples.

Although the manuscript is still lacking in discussion of the underlying physics that leads to the exceptional ZT, I believe the revisions made it suitable to be published on Nature Communications.

Reviewer #3 (Remarks to the Author)

The diligence of the authors to increase the quality of the manuscript is highly appreciated. I am still quite doubtful about the ability to n-dope SnSe, and would feel more comfortable if the authors could cycle the material in temperature a few times in order to make sure that they have not reached an un-reproducible metastable state.

Thus, I would be in favor of accepting this manuscript once the reproducibility of the results has been proven.

Reviewer #3: Comment:

“The diligence of the authors to increase the quality of the manuscript is highly appreciated. I am still quite doubtful about the ability to n-dope SnSe, and would feel more comfortable if the authors could cycle the material in temperature a few times in order to make sure that they have not reached an un-reproducible metastable state. Thus, I would be in favor of accepting this manuscript once the reproducibility of the results has been proven.”

Answer:

We appreciate very much on the referee’s comment. Indeed, the stability of thermoelectric materials at high temperature is a critical issue for applications. As the referee suggested, we investigated the reproducibility of transport properties for the highest ZT sample by conducting three continuous thermal cycles in the temperature range of 323 ~ 773 K. As shown in the below figures, all data present very consistent results each other and also agree with the previous result (black solid square in the figures) in the manuscript. Thus, the reproducible results indicate the stability of our n-type SnSe single crystals at high temperature. Power factors and ZT values also exhibit high reproducibility as shown in below figure c and d, respectively.

We added these data in the supplementary information.

<Figure r1. Temperature dependent (a) Seebeck coefficients, (b) electrical conductivities, (c) thermoelectric power factors, and (d) ZT of Bi-doped SnSe sample with carrier concentration of $2.1 \times 10^{19} \text{ cm}^{-3}$ at 773 K along b crystal axis. We performed the transport measurements with three continuous thermal cycles in the temperature range of 323 ~ 773 K. Old indicates the data in the manuscript for comparison.>

Reviewer #3 (Remarks to the Author)

The manuscript appears now acceptable for publication.